# Ciliary Signalling and Mechanotransduction in the Pathophysiology of Craniosynostosis

**DOI:** 10.3390/genes12071073

**Published:** 2021-07-14

**Authors:** Federica Tiberio, Ornella Parolini, Wanda Lattanzi

**Affiliations:** 1Dipartimento Scienze della Vita e Sanità Pubblica, Università Cattolica del Sacro Cuore, 00168 Rome, Italy; federica.tiberio@unicatt.it (F.T.); ornella.parolini@unicatt.it (O.P.); 2Fondazione Policlinico Universitario Agostino Gemelli IRCCS, 00168 Rome, Italy

**Keywords:** craniosynostosis, primary cilium, mechanotransduction, ciliopathies, suture ossification, craniofacial malformations, mesenchymal stromal cells, osteogenic pathways

## Abstract

Craniosynostosis (CS) is the second most prevalent inborn craniofacial malformation; it results from the premature fusion of cranial sutures and leads to dimorphisms of variable severity. CS is clinically heterogeneous, as it can be either a sporadic isolated defect, more frequently, or part of a syndromic phenotype with mendelian inheritance. The genetic basis of CS is also extremely heterogeneous, with nearly a hundred genes associated so far, mostly mutated in syndromic forms. Several genes can be categorised within partially overlapping pathways, including those causing defects of the primary cilium. The primary cilium is a cellular antenna serving as a signalling hub implicated in mechanotransduction, housing key molecular signals expressed on the ciliary membrane and in the cilioplasm. This mechanical property mediated by the primary cilium may also represent a cue to understand the pathophysiology of non-syndromic CS. In this review, we aimed to highlight the implication of the primary cilium components and active signalling in CS pathophysiology, dissecting their biological functions in craniofacial development and in suture biomechanics. Through an in-depth revision of the literature and computational annotation of disease-associated genes we categorised 18 ciliary genes involved in CS aetiology. Interestingly, a prevalent implication of midline sutures is observed in CS ciliopathies, possibly explained by the specific neural crest origin of the frontal bone.

## 1. Introduction

Craniosynostosis (CS) is the second most prevalent congenital malformation affecting the craniofacial complex after cleft lip/palate, with a prevalence that ranges between 1 in 1400 and 1 in 2100 [1]. CS results from the premature fusion of one or more calvarial sutures, the elastic fibrous tissue bands that connect the flat bones until skull growth is completed. Consequently, in CS the affected bones are prematurely fused hence less flexible, leading to a skull growth constraint (craniostenosis). A compensatory skull growth, sustained by the rapid brain development underneath, is increased along a direction that runs parallel to the closed suture (the so-called “Virchow’s law”), resulting in dysmorphisms of variable severity. 

The entire CS disease spectrum groups highly heterogeneous congenital defects, including primary CS—primitive genetic or idiopathic disorders—and secondary CS—consequences of other conditions affecting bone development, during the perinatal and postnatal growth stages (e.g., maternal hyperthyroidism, hypercalcemia, or rickets, and/or neurodevelopmental abnormalities of the foetus, such as microcephaly).

The alternative pattern of suture ossification allows categorising simple (single suture) versus complex (multi-suture) forms; while the occurrence of additional birth defects and/or developmental delay associated with CS allows denoting syndromic forms, rare diseases with wide clinical and genetic heterogeneity. Conversely, non-syndromic forms (NCS), which represent over 75% of primary CS cases worldwide, usually occur as isolated and single suture defects [2,3,4,5,6].

The rapid development of molecular genetic diagnostics and, in particular, the advent of high-throughput next generation sequencing technologies, have allowed a rapid improvement in the understanding of CS genetic aetiology, over the past two decades. To date, a growing number of genes can be listed as causative and/or associated to CS, including those found mutated in very rare diseases. Three quarters of all CS-associated genes are found mutated in syndromic forms, whereas a limited number of genes have been rarely associated with non-syndromic phenotypes [2,7]. Appendix A shows a quite exhaustive list of the known CS-associated genes, including those annotated in the OMIM database and those reported in the recent PubMed literature as associated with syndromes encountering CS in their phenotype.

Taking into account the wide genetic heterogeneity of the disease, Twigg and Wilkie provided an interesting overview of the main pathogenic mechanisms involved in cranial suture biology and development, considering the known disease-associated genes. This enabled classifying a discrete set of functional modules within an ontogenetic framework, relating to different stages of cranial suture development and inferring genotype/phenotype correlations [7]. Indeed, several genes can be categorised within a number of partially overlapping pathways, which allow defining the corresponding CS syndromes as pathway disorders, such as Fibroblast Growth Factor Receptor (FGFR) syndromes, RASopathies, Transforming Growth Factor β (TGFβ)-related syndromes, and Hedgehog (HH) signalling-related developmental disorders, among others. Within this genetic heterogeneity, a specific subset of CS syndromes can be classified as “craniofacial ciliopathies”, resulting from defects in primary cilia.

In this review we aim to specifically highlight the implication of genes orchestrating the primary cilium structure and function in the pathophysiology of CS, by dissecting their biological functions during craniofacial development and dysmorphogenesis.

## 2. Primary Cilium: A Sensing Organelle and a Signalling Hub

Since the original observation of a single non-motile cilium detectable on the surface of most vertebrate cells [8], this structure has been long disregarded as a vestigial organelle without a defined function. Significant scientific evidence overcame this outdated hypothesis, suggesting the functional implication of the primary cilium, expressed almost ubiquitously in vertebrate cells, in a variety of key processes.

It is currently established that the primary cilium acts as a cellular antenna receiving physical and biochemical extracellular stimuli and transducing them into intracellular signalling to regulate different processes during development and tissue homeostasis [9,10,11]. 

### 2.1. Structural Overview

The primary cilium has a microtubule-based axoneme core extending from the basal body, which arises from the mother centriole of the centrosome (see a scheme in Figure 1) [12,13]. The axoneme extends from the apical surface of the basal body as a cylindrical array of nine outer microtubule doublets projecting from cell body. The axoneme of the primary cilium displays a “9 + 0” array, as it lacks the central pair of microtubules and dynein arms, observed in motile cilia [14,15].

In dividing cells, the same centrioles give rise to the mitotic spindle fibres. Therefore, the primary cilium is a dynamic organelle that assembles when cells are in interphase and resorbs prior to mitosis to release the centriole, through the action of key cell cycle regulators (e.g., mitotic kinase Aurora A, polo-like kinase 1, Dishevelled 2, calcium/calmodulin signalling, and NIMA-related kinase 2) [16,17,18].

Ciliogenesis begins with the migration of the mother centriole equipped with appendages at the apical cell surface to become the basal body [19,20,21,22]. The basal body-derived centriole remains located in a depression of the plasma membrane named the “ciliary pocket” [23]. Here, transition fibres link the basal body microtubules to the ciliary membrane, constituting a gate for the docking of vesicles that carrycargoes and transmembrane proteins (Figure 1) [24].

Numerous proteins are associated with the basal body and contribute to the biogenesis, maintenance and function of the primary cilium. These include the BBSome complex (Figure 1), a stable heptameric structure consisting of eight subunits, namely, the Bardet Biedl Syndrome (BBS)-associated proteins and interacting proteins (BBS1, BBS2, BBS4, BBS5, BBS7, BBS8, BBS9 and BBIP10/BBS18/BBIP1) [25,26,27]. 

The BBSome contributes to the migration of the mother centriole to form the basal body owing to specific interactions among BBS subunits and centrosomal proteins. In addition, this complex contributes to the assembly of the intraflagellar transport (IFT) particles and attends the bidirectional transport of ciliary cargoes, by providing adaptor proteins for IFT complexes, enabling both vesicular trafficking towards the base of the cilium and the regulation of ciliary transmembrane protein composition [28,29,30,31,32] (Figure 1). 

The basal body is surrounded by the transition zone (TZ), where microtubule triplets of the basal body convert into microtubule doublets of the axoneme [33,34]. The TZ is characterised by Y-shaped links connecting the axonemal microtubules to the ciliary membrane (Figure 1). The transition fibres of the TZ play an important role in ciliary compartmentalisation constituting a selective barrier at the base of cilium to control ciliary trafficking of soluble lipids and proteins, and their selective import/export from the cilium [35].

The ciliary compartment is enclosed by a specialised membrane, continuous with the plasma membrane, enriched with a unique lipid and protein composition [36,37]. These include a variety of receptors, ion channels and transporters, enabling to sense and transduce intracellularly physical and chemical stimuli, such as mechanical loading (cilium deflection) or specific ligands (growth factors, hormones and morphogens) [38,39].

Protein trafficking towards and along the primary cilium is accomplished by a specialised transport system known as intraflagellar transport (IFT), which is mediated by microtubule-associated motor proteins (dyneins and kinesins) and IFT particles (Figure 1) [40,41,42]. Functional studies have shown that IFT is essential for assembly, maintenance and function of the primary cilium. The IFT particles include the IFT-A complex, which mediates the retrograde transport (from the tip to the basal body) by the dynein-2 motor protein, and IFT-B complex, which mediates anterograde transport by the kinesin-2 motor protein. Both complexes comprise several IFT proteins that assemble in subcomplexes and transiently bind to soluble proteins and tubulin monomers, to mediate their bidirectional transport along the organelle (Figure 1) [43,44]. The anterograde IFT promotes the trafficking of αβ-tubulin dimers, contributing to axonemal microtubule elongation, while the retrograde transport mainly mediates the import of membrane proteins across the ciliary gate at the transition zone [45,46].

### 2.2. Primary Cilium Signalling and Its Role in Craniofacial Development

The primary cilium transduces extracellular signals as it houses an intense signalling, mediated by different pathways, of which Hedgehog (HH) and Wingless-related integration site (Wnt), play pivotal roles [9,37].

The HH pathway is one of the most important cilium-related signalling, regulating different processes during embryonic development and tissue homeostasis [37,38]. In the absence of HH ligands, the 12-transmembrane receptor patched 1 (PTCH1) locates in the ciliary membrane and constitutively inhibits the ciliary localisation of the G protein-coupled receptor Smoothened (SMO) (Figure 1) [47]. The translocation of HH molecules inside and outside the cilium compartment is facilitated by the IFT complexes [48,49]. The binding of ligands, such as Sonic Hedgehog (SHH) or Indian Hedgehog (IHH), to PTCH1 causes the displacement of the receptor from the cilium, through endocytic degradation by the SMAD-specific E3 Ubiquitin Protein Ligase 1 and 2 (SMURF1 and 2) and allows the translocation of SMO in the ciliary membrane. In the absence of HH ligand, SMO is inhibited by PTCH1; whereas, upon HH ligand binding and PTCH1 degradation, SMO becomes phosphorylated by casein kinase 1 (CK1) and G-protein coupled receptor kinase 2 (GRK2), moves into the primary cilium (PC), and assumes an activated conformation [50,51]. Once activated, SMO promotes the dissociation of GLI transcription factors from the negative regulator, suppressor of fused (SuFu), causing their activation and nuclear translocation to regulate target genes’ transcription. GLI3, in particular, is a C2H2-type zinc finger transcription factor, with a dual activity within the HH pathway and plays a key role in vertebrate development. With active HH signalling the full length GLI3 protein undergoes phosphorylation and nuclear translocation, and acts as a transcriptional activator (GLI3-A) on target gene expression. Conversely, when the HH is off, GLI3 is processed by C-terminus truncation into the GLI3-R transcriptional repressor [52] (Figure 1).

Wnt signalling is instead initiated by the binding of Wnt ligands to the receptor Frizzled (Fz) associated with the Low-density lipoprotein-related receptors 5 and 6 (LRP5/LRP6) and Dishevelled (Dvl) [53]. In the absence of Wnt ligands, the cytosolic-β-catenin is targeted for proteasomal degradation by the Axin/Adenomatous Polyposis Coli (APC)/Glycogen synthase kinase 3β (GSK3-β) complex. The binding of Wnt ligands to its receptor Fz allows the recruitment of Axin and promotes the stabilisation of β-catenin and its nuclear translocation. In the nucleus, β-catenin induces the expression of target genes involved in cells proliferation, differentiation and survival/apoptosis [54,55].

While the role of the primary cilium in canonical Wnt signalling is still controversial, it is essential for the planar cell polarity (PCP) regulated through the β-catenin-independent non-canonical Wnt signalling [56]. Indeed, the migration of the mother centriole to the apical cell surface to form the basal body defines apical-basal polarity of the cell, a process coordinated by proteins of the ciliary TZ [9,57,58,59]. 

The primary cilium has been identified ex vivo on the cell surface of craniofacial tissues and strongly contributes to the craniofacial development driven by a complex circuitry of molecular signals, including HH, Fibroblast Growth Factor (FGF), TGF-β/bone morphogenetic protein, and Wnt pathways, among others [60,61]. Craniofacial development is a complex multi-step process to which all the three germ layers contribute, exchanging reciprocal inductive signals, to enable the synchronous growth of the brain, of the braincase and of the surrounding tissues and organs. The cranial vault, in particular, forms through the progressive direct (membranous) ossification of dermal bone plates, thanks to the condensation and osteogenic differentiation of mesenchymal progenitors, induced by both biochemical and mechanical stimuli from the extracellular environment [62]. The mesenchymal stromal cells (MSC) that give rise to the skull vault derive from two alternative depots with different germ layer origins: the neural crest, originating from the ectoderm through epithelial–mesenchymal transition, and the paraxial mesoderm, segregating as an unsegmented portion to form the head mesoderm. The two MSC types have different topographical distributions in the newly formed skull: neural crest-derived cells (NCCs) will form the frontal bone and the entire facial skeleton, while the paraxial mesoderm will form the rest of the neurocranium, which has indeed an entirely mesodermal origin [63]. This way, the cranium develops, in all vertebrate species, by integrating genetic inputs and mechanical forces from growing soft tissues to determine skull morphology [64]. In these processes, the primary cilium serves as a signalling hub and a mechanotransduction relay among the forebrain neuroectoderm, the surface ectoderm, the endoderm and cranial neural crest cells.

In this developmental context, HH signalling is essential for mesodermal tissue patterning and differentiation, thereafter, in later stages, it is also needed for cranial suture morphogenesis and intramembranous ossification [65,66]. Indeed, the dysregulation of this pathway results in a wide array of craniofacial developmental defects [67]. In particular, SHH is involved in the majority of HH function during face and skull morphogenetic processes, including the first pharyngeal arch formation, the survival and the proliferation of cranial neural crest cells, the development of the maxilla, of the palatal clefting, and of the mandible and the fusion of craniofacial prominences [65,68,69]. IHH acts by regulating bone development [70,71]. Studies in mouse mutants defective for primary cilium components, confirmed the essential role HH signalling during craniofacial development [48,72]. 

The canonical Wnt signalling also acts in many aspects of skeletal development, including skull morphogenesis. Wnt activation is required for NCCs’ migration, proliferation and osteoblast differentiation [73,74]. A mutant mouse model expressing a constitutively active canonical Wnt signalling with stabilised β-catenin features reduced skull bone formation [75]. 

The role of Wnt pathway during development widely depends upon functional primary cilium processes [76,77,78]. Liu et al. revealed that NCCs with deletion of *Kif3a* and defective primary cilium showed an aberrant activation of the Wnt pathway. This effect appeared to be context-related, as it was observed only in domains where NCCs receive Wnt signals from the surrounding environment, requiring a functional primary cilium [79].

## 3. Craniosynostosis as a Primary Cilium Defect: Ciliary Genes in Syndromic CS

The abnormal development of the craniofacial complex and/or of the entire skeleton, often occur in a number of ciliopathies, including the Bardet–Biedl syndrome (OMIM #209900), the Meckel–Gruber syndrome (OMIM #249000), the Joubert syndrome (OMIM #213300), the oral-facial-digital syndromes (OMIM #311200), the short rib-polydactyly syndromes (OMIM #613091, #263520), the Jeune asphyxiating thoracic dystrophy (OMIM #208500), the Sensenbrenner syndrome (OMIM #218330), and the Ellis–van Creveld syndrome (OMIM #225500) [60]. In particular, CS may represent a relevant feature in syndromes associated with mutations in ciliary associated genes. Observing the entire scenario of CS genetics (see Appendix A), discrete set of genes can be categorised within a number of defined and partially overlapping pathways, according to gene ontology (GO) annotations (see Appendix A for complete details on GO annotations). 

In particular, 18 out of 91 genes, listed in Table 1, are implicated in the assembly, function and regulation of the primary cilium, based on the information available in the ciliome database [80] and on the GO ‘biological process’ and ‘cellular component’ annotations (Appendix A). 

All but one of these genes are functionally interconnected at multiple levels, as shown in Figure 2. In the following paragraphs, we attempted a genotype/phenotype correlation, discussing the functional implications of these ciliary genes categorised based on the associated syndromes and disease spectra.

### 3.1. Cranioectodermal Dysplasia Disease Spectrum

Genes encoding IFT components are implicated in the etiopathogenesis of cranioectodermal dysplasia (CED), also known as Sensenbrenner syndrome, an autosomal recessive ciliopathy with skeletal involvement. CED is genetically heterogeneous, with homozygous or compound heterozygous mutations found in either of *IFT122* (mutated in classical CED phenotype), *WDR35/IFT121* (associated with CED2), *IFT43* (causing CED3), or *WD Repeat Domain 19 (WDR19)/IFT144* (causing CED4) genes. All CED-associated genes encode parts of the core and peripheral subcomplexes of the IFT-A particle (see Section 2.1 and Figure 2). They play leading roles in ciliogenesis, and in the translocation of GPCRs to cilia [85,86,87,88].

The CED phenotype is characterised by sagittal (sometimes also metopic) CS plus thorax and limb abnormalities (narrow thorax, shortened proximal limbs, syndactyly, polydactyly, brachydactyly, joint laxity), along with growth deficiency, ectodermal (widely spaced hypoplastic teeth, hypodontia, sparse hair, skin laxity, abnormal nails) and facial features (high forehead with frontal bossing, low-set ears, telecanthus, epicanthal folds, full cheeks, everted lower lip). Visceral anomalies (nephronophthisis, hapatic fibrosis) are also observed and represent major causes of morbidity and mortality [89,90,91].

**Figure 2 genes-12-01073-f002:**
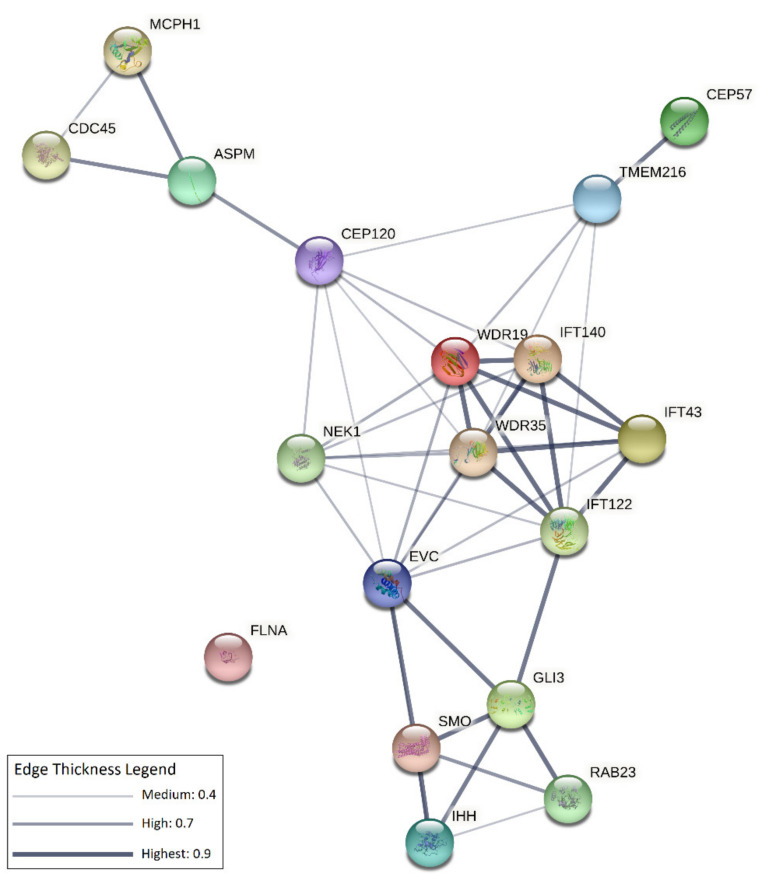
Network of interactions among CS-associated ciliary genes. The network was obtained using the STRING (v.11.0) software (https://string-db.org/, accessed on 8 July 2021), which assigns scores, and integrates protein–protein interaction data and complements these with computational predictions [92]. Each node in the network represents a protein/protein-coding gene, edges represent protein–protein interactions that can be either derived from curated databases, or experimentally determined, or based on computational predictions (gene neighbourhood, gene fusions, gene co-occurrence), or derive from alternative publicly available sources of information (text-mining, co-expression, protein homology). All genes implicated in ciliary functions, listed in Table 1, were used as inputs for building the network. The resulting network contains 18 interconnected nodes. The line thickness indicates the strength of data support (see the legend in the upper right side of the figure and refer to String version 11.0 https://string-db.org/ (accessed on 8 July 20) for additional details).

### 3.2. Short-Rib Thoracic Dysplasia

Short-rib thoracic dysplasia (SRTD) refers to another group of autosomal recessive skeletal ciliopathies, showing both clinical and genetic overlaps with CED. SRTD show a wide genetic heterogeneity, with 20 different coded phenotypes associated with different loci annotated in the OMIM database, to date. The genetic aetiology includes compound heterozygous mutations of *WDR35* and of another IFT encoding gene, namely, *IFT140*. More specifically, the phenotype associated with *IFT140* mutations (SRTD9), was recently reported to include sagittal and metopic CS as part of the observed skeletal malformations [93]. 

In addition, biallelic mutations of the centrosomal protein 120 gene (*CEP120*), already associated with Joubert syndrome, cause a variety of SRTD, also known as Jeune asphyxiating thoracic dystrophy. The phenotype is characterised by a long, narrow thorax, short-limbed short stature, polydactyly, and renal cystic dysplasia. Unilateral coronal craniosynostosis was reported in one affected family, broadening the phenotypic spectrum [94]. Finally, the NIMA-related kinase 1 gene (*NEK1*) is another disease gene for a SRTD phenotype (SRTD6), which may feature bicoronal synostosis [95]. Interestingly, both CEP120 and NEK1 are involved in centrosome organisation and dynamics and play pivotal roles in centriole biogenesis and cell cycle progression control. CEP120 is necessary for microtubule-dependent coupling of the nucleus and the centrosome and in centriole elongation during ciliogenesis, interacting with other centriole assembly proteins [96]. NEK1 is instead a serine-threonine kinase, member of the NIMA-related kinases family, involved in centrosome disjunction and cilium assembly, along with additional pleiotropic functions in cell cycle checkpoint control and DNA damage response, in vertebrate cells [97,98].

### 3.3. HH Signalling-Related Syndromes

Genes implicated in the HH pathway cause different syndromic CS, classified as ciliopathies. These include, Joubert syndrome spectrum, typical ciliopathies with severe neurodevelopmental disorder, eye and kidney abnormalities, and a multiorgan involvement, has been extended to include midline CS, defining the Joubert syndrome 2. This is associated with homozygous mutations in the gene encoding the transmembrane protein 216 (*TMEM216*). TMEM216 is a membrane protein expressed at the ciliary base-TZ, as part of the tectonic-like complex, which regulates the HH pathway, is required for tissue-specific ciliogenesis and regulates ciliary membrane composition [99].

In the context of HH signalling, microduplications of the *IHH* gene locus on 2q35 have been found to segregate with the Syndactyly type 1 phenotype, unrelated multigeneration kindreds, featuring sagittal CS [100,101]. The critical duplicated region included a regulatory sequence upstream the gene, that was predicted to serve as a long-range enhancer of *IHH*, regulating its expression during bone formation, hence affecting digit and skull development [100]. 

A somatic mosaic mutation in the SMO receptor gene causes the Curry–Jones syndrome, a multisystem disorder characterised by brain malformations, unicoronal craniosynostosis, patchy skin lesions, polysyndactyly, iris colobomas, microphthalmia, and intestinal malrotation with myofibromas or hamartomas [102]. 

Heterozygous loss-of-function mutations of *GLI3* cause the Greig Cephalopolysyndactyly syndrome, with a widely variable expressivity, in which digital malformations are associated with sagittal and metopic CS [103]. 

Finally, homozygous mutations of the *RAB23* gene cause the Carpenter Syndrome 1, a rare pleiotropic disorder featuring either simple or multi-suture synostosis as a hallmark, within a multisystem developmental impairment, with variable severity, affecting the brain, the genitourinary system, the heart and the appendicular skeleton [104]. RAB23 is a small GTPase superfamily member, acting as a vesicular transport protein that regulates endosome formation and vesicular trafficking during the receptor-mediated endocytosis activated by HH binding to PTCH1, hence serving as a negative regulator of the HH signalling [105,106]. In addition, RAB23 regulates suture biogenesis and osteoprogenitors’ commitment by repressing the FGF-ERK1/2 signalling [107]. The CS in this syndrome may vary from simple sagittal CS, to combined midline (sagittal and metopic) CS, to complex involvement including cloverleaf skull [108,109]. Interestingly, the heterogeneous clinical spectrum of this syndrome may include developmental abnormalities that are typically recognised as ciliopathy traits, including *situs inversus totalis* [110].

To date, a single case of sagittal synostosis has been reported in a patient affected by Ellis–van Creveld syndrome, an osteo-chondral dysplasia mainly affecting limbs, associated with cardiac defects [83]. Homozygous mutations in either the *EVC* or the *EVC2* genes have been usually described in this disorder [111]. The Ellis–van Creveld syndrome gene (*EVC*) encodes a positive regulator downstream the HH signalling, expressed on the ciliary membrane as a single-pass transmembrane protein [112]. EVC binds the homologous EVC2 (EVC ciliary complex subunit 2) to form a heterodimeric complex that interacts with SMO and controls HH pathway activity by regulating Sufu/GLI3 dissociation and GLI3 trafficking in primary cilia [113].

### 3.4. Other Rare Syndromes and Phenotypes

Craniofacial dysmorphisms, including scaphocephaly, were associated with the mosaic variegated aneuploidy syndrome, caused by biallelic *CEP57* mutations. This autosomal recessive disorder is characterised by poor growth and congenital heart defects, linked to mosaic aneuploidies, explained by the role of centrosome in mitotic spindle assembly [114]. CEP57 is another centrosomal protein, which is specifically required for microtubule attachment to centrosomes and also for the nuclear translocation of mitotic checkpoint regulators [115].

An association with CS was also described in microcephaly 1 syndrome, caused by homozygous mutations in the microcephalin 1 *MCPH1* gene. The CS in this condition mainly affects the metopic suture resulting in the typical “bird-like facies” of patients [116,117]. Additionally, the *CDC45L*, encoding the essential component of the DNA replisome, CDC45, is associated with a microcephaly phenotype in which CS of variable severity, ranging from unilateral or bilateral coronal synostosis to multiple suture involvement, has been found [118,119]. CDC45 and MCPH1 are instead nuclear proteins involved in cell cycle control, acting on the G1/S and G2/M checkpoints, respectively [119,120]. Metopic CS was recently reported in two siblings with a compound heterozygous mutation of the abnormal spindle homolog of microcephalin (*ASPM*), who showed a milder phenotype of microcephaly 5, featuring normal early development [81]. ASPM is mitotic spindle protein that associates with centrosome and axoneme, through ASPM-SPD-2-Hydin (ASH) domains, playing a key role in mitotic spindle formation [121,122].

Furthermore, sagittal CS represents a feature in gain-of-function mutations in the *FLNA* gene on chromosome Xq28, which cause a phenotype included in the otopalatodigital syndrome spectrum, characterised by skeletal anomalies, and severe malformations in the hindbrain, heart, intestines, and kidneys, with high risk of perinatal death [84,123]. *FLNA* encodes the actin-binding protein filamin A, expressed at the basal body of the primary cilium, where it interacts with meckelin to mediate basal body positioning and ciliogenesis and it regulates the levels of Wnt signalling [124].

A comprehensive overview of the ciliary genes implicated in CS phenotypes, clustered according to their function and localisation in ciliary structures, is shown in Figure 3.

## 4. The Primary Cilium in Suture Niche Biology: Mechanotransduction in the Etiopathogenesis of Non-Syndromic Craniosynostosis

The growth and patterning of craniofacial sutures are widely affected by mechanical forces, through mechanotransduction mechanisms occurring at the boundaries of cranial vault bones, whose dysregulation may explain most of the pathophysiology of craniosynostosis [125,126]. 

Cranial sutures are fibrous joints separating flat calvarial bones, composed of two osteogenic fronts and an intervening mesenchyme-derived fibrous tissue [127,128]. The inter-suture mesenchyme constitutes a unique skeletal niche containing MSC, which represent a transient reservoir of new progenitors contributing to preserve suture patency, and disappearing upon completion of suture ossification [129]. MSC differentiate into osteogenic cells promoting suture ossification during childhood and adulthood [130,131,132]. The behaviour of these cells is influenced by biochemical and physical factors, including mechanical forces [133,134].

During normal development, patent sutures allow the expansion of the skull in order to support the brain growth, being flexible structures able to resist to tensile or compressive forces [125]. In addition, sutures allow deformation of the skull during childbirth, absorb cyclic mechanical loading during mastication and locomotion, and act as shock absorbers against internal (blood vessel pulsation, intracranial pressure, forces implemented by the dura mater) or external forces [128,135]. These biomechanical stimuli are translated into biological signals that result in the proliferation or in the osteogenic differentiation of cells within the sutures [136,137].

Since ossification of skull sutures is also mediated by biophysical mechanisms from the surrounding environment, premature fusion of the sutures in CS patients has been linked to alteration of mechanical stimuli applied to the calvarial niche, or to an aberrant perception and response of MSC to these stimuli [134,138,139,140]. On this regard, in vivo studies on mouse CS models and epidemiological analyses have shown that foetal head constraint caused by early descent in pelvis, primiparity, twin or multiple pregnancies, high birth weight, and late term pregnancy was associated with an increased risk to develop non-syndromic CS [126,141,142,143].

It has been widely demonstrated that mechanical forces applied to the suture mesenchyme induce MSC activation and commitment, resulting in changes in cell morphology, size and number, vascularisation, suture morphology and in increased expression of osteogenic genes [144,145]. For example, an in vitro study showed that cyclical compressive loads applied on a sagittal suture model caused premature ossification, due to the underlying upregulation of osteogenic markers during the application of the mechanical stimuli [136]. More recently, Barreto and co-workers analysed the effect of microenvironmental stiffness on the osteogenic commitment of cells isolated from CS patients’ sutures [133]. This study enabled identifying the gene expression profiles associated with the cellular mechano-response, including the upregulation of genes mediating bone growth and development, extracellular matrix remodelling, inflammation and osteogenic differentiation [133]. 

Several studies have recently illustrated the emerging mechanosensory role of the primary cilium also in bone formation [144,146,147,148,149,150,151]. Indeed, the primary cilium protruding from the apical surface of osteogenic precursors and bone cells, is able, through specific receptors and mechanosensitive channels, to perceive osteoinductive stimuli and transduce them to activate the intracellular osteogenic cascade [152,153,154]. On this regard, mice with an osteoblast- and osteocyte-specific knockout of *Kif3a* exhibited reduced bone deposition in response to mechanical ulnar loading compared to control [155]. Kif3a is a subunit of the motor protein kinesin II that is involved in the IFT-B-mediated transport (Figure 1), hence contributing to the formation, function and maintenance of the primary cilium. 

A study led by Chen et al. showed that bone marrow cells transplanted in a murine model were recruited within the bone-forming surfaces in response to loading, demonstrating that mechanical signals enhance the homing and differentiation of osteogenic precursors to bone surfaces. Disruption of the mechanosensing organelle in transplanted osteoprogenitors significantly decreased the bone matrix deposition confirming the essential role of the primary cilium in this process [146]. 

The involvement of the primary cilium in osteogenic mechanosensing has been also proposed in non-syndromic sagittal CS [156]. Mesenchymal stromal cells isolated from prematurely fused sutures of non-syndromic patients displayed a dysregulated expression of a splice variant of BBS9, a protein of the BBsome complex, resulting in a reduced tendency to form functional primary cilia upon osteogenic induction compared with control cells. This suggests that the pathogenic events leading to premature suture fusion involve the reduced primary cilium expression on osteogenic precursors, which are thus unable to properly sense and respond to osteogenic stimuli from the surrounding environment. This causes the progressive exhaustion of the mitotic MSC reservoir in the suture niche mesenchyme [130,156]. Interestingly, part of the upregulated gene signature observed in fused sutures of non-syndromic CS patients [156] overlaps with the stiffness-induced gene expression profile reported by Barreto and co-workers [133], thus further supporting the stiffness-related pathophysiology of CS.

A fine dissection of the molecular mechanisms implicated in the ciliary mechanotransduction involved in bone formation derives from fluid flow biomechanics studies in cells. Human MSC respond to oscillatory fluid flow by proliferating, activating osteogenic signalling cascades, hence undergoing osteogenic differentiation [144,157,158]. Additionally, dynamic fluid flow results osteocyte-mediated inhibition of osteoclastogenesis by increasing the OPG/RANKL ratio [148]. Interestingly, fluid shear-related osteogenic induction inversely correlates with the activation of HH signalling, as demonstrated by the reduced expression of PTCH1 and GLI1 [144,159]. 

A main role in cilium-related fluid mechanosensing is played by the complex formed by polycystin-1 (PC1) and polycystin-2 (PC2), which function as a mechano-induced membrane receptor and a calcium-permeable ion channel, respectively, in the primary cilium [160]. The polycystin complex plays an essential role in kidney and liver by sensing and responding to flow-induced shear stress. Mutations in the polycystic kidney disease genes 1 and 2 (*PKD1-2*), encoding PC1-2, cause autosomal dominant polycystic kidney disease. Their discovery was a milestone in primary cilium biology research, as the importance of this organelle in human diseases was first recognised when the ciliary localisation of PC1-2 proteins was originally detected [60]. Polycystin-related mechanotransduction plays a role also in bone development; indeed, PC1 and PC2 localise to the primary cilium in bone cells. Short term mechanical stretching applied on osteoblast progenitors induces a PC1-mediated osteogenic cascade, involving activation of the Runt-related transcription factor 2 (RUNX2) and of the extracellular signal-regulated kinase (ERK) signalling [137,161]. These in vitro observations are further supported by in vivo evidence: heterozygous Pkd1 mutant mice have decreased bone mineral density and a reduced expression of the osteogenic genes [162,163].

## 5. Consideration on Suture Site Involvement

Despite the wide phenotypic variability within the entire CS spectrum, a general observation can be made on suture involvement. The sagittal suture tends to be prevalently involved, both in syndromic phenotypes associated with gene mutations in ciliary genes, and in non-syndromic sporadic cases plausibly influenced by mechanotransduction as an environmental risk factor. 

The sagittal suture is affected in 11 out of the 18 (61%) mendelian syndromes discussed in paragraph 3 (see Table 1), whereas in two additional cases, both the sagittal and the metopic suture can be affected by the dysmorphism. Therefore, overall the midline sutures are the involved in 72% of the syndromic phenotypes caused by mutations in ciliary genes. The remainder of cases (5 of 18, 28%) involve the coronal suture, either unilaterally or bi-laterally, or multiple sutures.

The dual developmental path of skull sutures may explain these preferential suture sites, at least in part: the sagittal and metopic sutures derive from the neural crest, while the other cranial bone structures share a mesodermal origin. Given the crucial role of the primary cilium and of the molecular signalling that occur within the cilioplasm during early stages of embryo development, it is not surprising that mutations in ciliary genes may affect the correct development of the primordium of the skull, involving neural crest cell-derived structures, rather than later craniofacial patterning and morphogenetic events.

On the other hand, non-syndromic sagittal synostosis is regarded as the most prevalent CS variety and the less influenced by the genetic burden, suggesting environmental and/or epigenetic influences [1,164]. Foetal head constraint is thought to represent one of the possible environmental risk factors for non-syndromic sagittal synostosis, although yet to be demonstrated [140,141]. Nevertheless, a contribution of mechanotransduction and related primary cilium signalling in the pathophysiology of non-syndromic sagittal CS has been proposed in in vitro studies [156], discussed in Section 4. Indeed, the primary cilium acts as a cellular antenna, to bridge abnormal mechanical stimuli from the environment to defective developmental paths due to the underlying genetic background.

## 6. Conclusions

The molecular pathophysiology underlying the widely heterogeneous spectrum of craniosynostosis clearly indicates contributions from both genetics and the environment in causing premature suture fusion. The role of the primary cilium emerges in this scenario as a key cellular compartment, as it houses molecular networks that include sensing molecules expressed on the ciliary membrane and intracellular messengers and effectors merged in the cilioplasm. An improved understanding of the role of the primary cilium in governing the healthy patterned growth of the calvarium, and of its influence on osteogenic cell proliferation, differentiation, and migration during development would enable shedding new light in the pathophysiology of both syndromic and non-syndromic craniosynostoses. The exhaustive knowledge of molecular mechanisms would in fact potentially prompt the discovery of novel disease-associated genes and of genetic risk factors, through a bottom-up approach based on evidence from the biology of suture patterning and regulation. 

## Figures and Tables

**Figure 1 genes-12-01073-f001:**
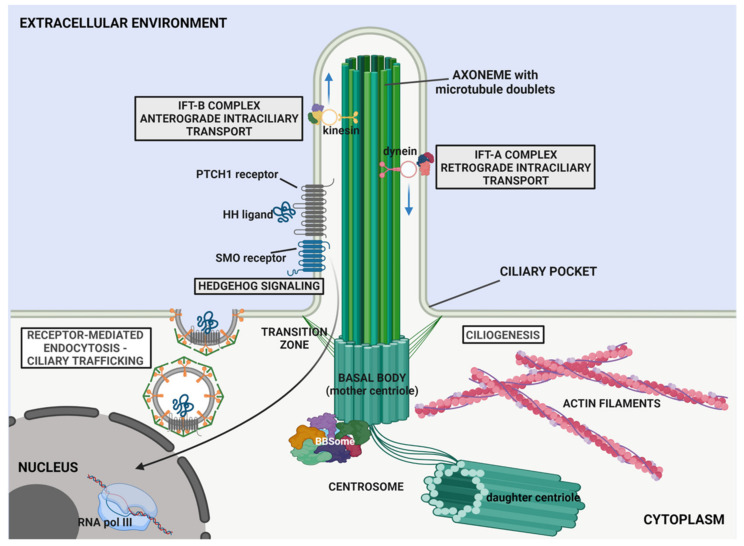
Schematic representation of the primary cilium structure. The primary cilium protrudes from the apical surface of cells, with a central shaft composed of nine microtubule doublets (axoneme) that extend from the basal body, a modified centrosomal mother centriole. The transition zone (TZ) is characterised by Y-shaped links connecting the axonemal microtubules to the ciliary membrane contributing to compartmentalisation of the organelle. Proteins and other cargos are transported from the basal body to the tip of axoneme by anterograde intraflagellar transport (IFT) through IFT-B complex and kinesin-motor protein; whereas dynein-motor protein associated with IFT-A complex contributes to retrograde IFT (from the tip to the basal body). The basal body associates with the BBSome, a heptameric complex involved in ciliogenesis and ciliary trafficking. The ciliary membrane contains specialised lipids, proteins and receptors (e.g., Patched 1 (PTCH1) receptor and the associated Smoothened (SMO) co-receptor, that bind different hedgehog, HH, ligands), through which the cilium coordinates different pathways (e.g., Hedgehog signalling). See text for further details.

**Figure 3 genes-12-01073-f003:**
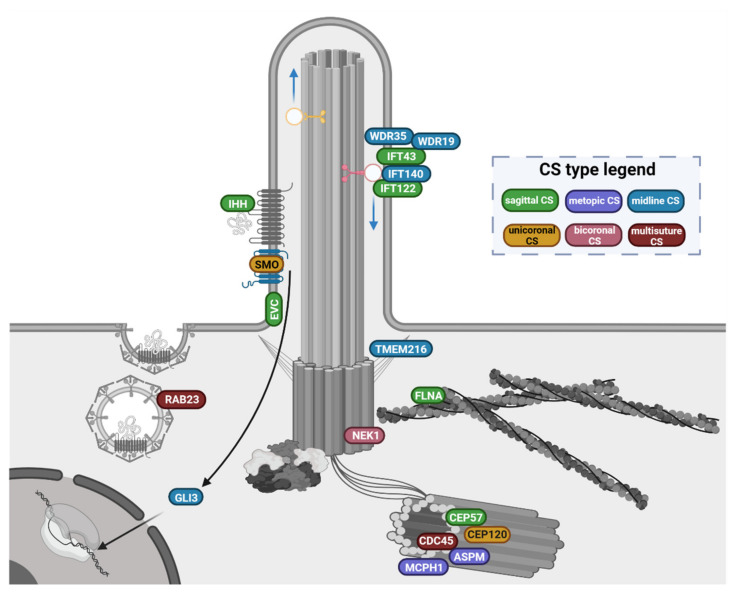
Overview of the primary cilium-related gene products involved in CS pathophysiology. Genes are clustered according to their function and location in the ciliary structures. Gene products labels are coloured after the suture sites (CS type legend) that are prevalently involved in the associated phenotypes (see Section 3.1 and Section 3.2 for details). The functional ciliary gene clusters and the associated phenotypes indicate some level of genotype/phenotype correlations: (i). the genes encoding adaptor proteins needed for the intraflagellar complex A-mediated retrograde transport (*IFT122, IFT43, IFT140, WDR35, WDR19*) are found mutated in phenotypes with midline CS, prevalently affecting the sagittal suture; (ii). genes encoding main players of the hedgehog pathway (*SMO, GLI3, IHH, EVC* and *RAB23*) are associated with more heterogeneous suture involvement, ranging from single sagittal or unicoronal CS, to bicoronal and multi-suture craniosynostoses; (iii). the genes products involved in the regulation of centrosome cycle, ciliary trafficking and cilium assembly are mutated in syndromes that may feature either midline CS (*ASPM, MCPH1, CEP57, TMEM216, FLNA*), or uni/bicoronal CS (*NEK1, CEP120*) up to multisuture CS (*CDC45*).

**Table 1 genes-12-01073-t001:** Ciliary genes implicated in craniosynostosis syndromes.

Gene Symbol	Associated Syndrome or Phenotype	Main Pathways/Molecular and Biological Functions	Involved Suture(s)	OMIM/PubMed Reference
*ASPM*	Microcephaly 5, Primary, Autosomal Recessive	Centrosome cycle; Cell cycle	Metopic	#608716, [81]
*CDC45*	Meier-Gorlin Syndrome (Atypical)	Cell cycle	Coronal	#224690, [82]
*CEP120*	Short-Rib Thoracic Dysplasia w/o Polydactyly	Cell cycle; Cilium biogenesis and maintenance	Coronal	#616300
*CEP57*	Mosaic Variegated Aneuploidy Syndrome 2	Cell cycle; Cilium biogenesis and maintenance	Sagittal	#614114
*EVC*	Ellis Van Creveld Syndrome	Hedgehog signalling pathway	Sagittal	#225500, [83]
*FLNA*	Otopalatodigital Spectrum Disorders	MAPK signalling pathway	Skull base/Multisuture	*300017, [84]
*GLI3*	Greig Cephalopolysyndactyly Syndrome	Hedgehog signalling pathway	Sagittal/Metopic	#175700
*IFT122*	Cranioectodermal Dysplasia 1	Ciliogenesis and/or cilium maintenance; Ciliary protein trafficking; SHH signalling; Signalling by GPCR	Sagittal	#218330
*IFT140*	Short-Rib Thoracic Dysplasia 9 w/o Polydactyly Syndrome	Ciliogenesis and/or cilium maintenance; Signalling by GPCR; Ciliary protein trafficking	Sagittal	#266920
*IFT43*	Cranioectodermal Dysplasia 3	Ciliogenesis and/or cilium maintenance; Signalling by GPCR; Ciliary protein trafficking	Sagittal	#614099
*IHH*	Syndactyly, Type 1, w/o CS	Hedgehog signalling pathway	Sagittal	#185900
*MCPH1*	Microcephaly 1, Primary, Autosomal Recessive	Cell cycle; Bone development	Variable	#251200
*NEK1*	Short-Rib Thoracic Dysplasia 6 w/o Polydactyly	Cilium assembly	Coronal	#263520
*RAB23*	Carpenter Syndrome 1	Cilium assembly	Sagittal/Lambdoid, coronal	#201000
*SMO*	Curry–Jones Syndrome	Hedgehog signalling pathway; Axon guidance; Basal cell carcinoma; Pathways in cancer	Sagittal/Metopic	#601707
*TMEM216*	Joubert Syndrome 2	Cilium assembly	Multisuture	#608091
*WDR19*	Cranioectodermal Dysplasia 4 Syndrome (Frontal Bossing)	Cilium biogenesis and maintenance; Intraflagellar transport; Hedgehog signalling	Sagittal	#614378
*WDR35*	Cranioectodermal Dysplasia 2 Syndrome	Cilium biogenesis and maintenance; Intraflagellar transport; Hedgehog signalling	Sagittal	#613610

* The asterisk before an entry number indicates a gene of known sequence.

## Data Availability

No new data were created or analyzed in this study. Data sharing is not applicable to this article.

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
