# Peer review of "Ciliary Signalling and Mechanotransduction in the Pathophysiology of Craniosynostosis"

_genes, 2021, doi:10.3390/genes12071073_

Round 1
Reviewer 1 Report
The goal of this review paper is to describe the role of the primary cilium, a structure found in most vertebrate cells, in the pathophysiology of craniosynostosis. Craniosynostoses are highly prevalent craniofacial birth defects. If left untreated, this birth defect result in distortion of skull shape, increased intracranial pressure and facial deformities; thus, the topic is relevant to the craniofacial biology and development field. However, there are many major points that need to be addressed.
Major points:
Even though the authors are able to partially convey the topic to the reader, the written language needs major work. There are serious problems with the English language throughout the paper. After reading the paper, I had the impression some portions were written out of a translator software or different portions were written by different people with distinct English proficiencies. There are problems with wording and grammar. Adverbs are used often but most of the time they are not required again reflecting literal translation from another language. The order of words in many sentences also reflect literal translation from another language. Paragraphs are also another major problem: the authors seem to add paragraphs randomly without following a logical reason; some paragraphs have only 2 sentences demonstrating they should not be a paragraph. I am intentionally not pointing the areas with problems in language because I would have to add 95% of the paper. Therefore, extensive editing of English language and style is required; ideally the paper should be completely rewritten.
The paper is also poorly organized in relation to its sections. Information is repeated over and over. A major example is section 3 (3.1 and 3.2). The information in these 2 sections is repetitive because the authors chose to divide based on genotype and phenotype. Instead, a better approach would be a single section in which each one of the 18 genes has its own subsection. Another example is section 2 (2.3 and 2.4). In section 2.3, the author described some but not all the signals in the primary cilia; then in section 2.4, the authors go back to some of the same signals described in section 2.3 (e.g., Hh), and add more signals (e.g., PDGF) that were omitted in section 2.3. A single section related to primary cilium signaling in craniofacial development is sufficient. As a rule of thumb: “less is better”. Repetitive information or information not relevant to the topic just distracts the reader from the real goal of the paper.
As the title indicates, the paper is supposed to describe the role of the primary cilium in the pathophysiology of craniosynostosis. However, the authors begin the paper with a very long table containing many genes that are not relevant to this topic. There are 128 genes, however only 18 genes are linked to ciliopathies and craniosynostosis. Table 1 takes around 7 pages and conveys mostly irrelevant information to the subject. Table 1 must be removed from the paper completely or added to supplemental data. Only the 18 genes should be described in a shorter table.
Regarding table 1, by adding so many genes, the authors ended up not including the most relevant piece of information: the reference for the case report. They added the OMIM number. However, while some OMIM descriptions will mention craniosynostosis (e.g., 603595), others do not mention the presence of premature suture closure in the specific birth defect (e.g., 136760). As a matter of fact, when checking the clinical features for frontonasal dysplasia 1, OMIM states there is no craniosynostosis (“Twigg et al. (2009) disagreed with this classification, citing differences in facial morphology, absence of craniosynostosis, and autosomal recessive inheritance.”). While this does not mean craniosynostosis was not reported elsewhere for such birth defects, it shows the authors are referencing a source that does not contain the information they are describing in the table. You cannot reference a reference that references other sources. You must reference the case report describing the presence of craniosynostosis in the birth defects. This is one more reason why this table should be removed from the paper. Only the 18 genes should be emphasized throughout the paper.
Related to references, I was able to catch some references that were incorrect. One major example is when the authors reference the embryonic origin of the frontal bones (lines 245-252). While they correctly say the frontal bones are derived from neural crest cells, reference 90 states otherwise. In fact, this is not the right reference: there is work done in mouse using the Wnt1-Cre and R26R lines showing the neural crest origin of this bone. This mistake shows the authors were not careful about checking the references thoroughly.
Regarding the Hedgehog signaling, the authors need to be careful about adding the word “Sonic” or the letter “S” when describing this pathway. For instance, in section 3.1, the authors are describing the link of genes (genotype) in craniosynostosis. In line 389-392, they say Shh 3 times. However, the Ihh pathway is the one acting in the calvaria/cranial sutures; not the Shh as the authors mention in lines 378-380, right before going back to Shh pathway. This shows poor understanding of the signaling pathways acting in the calvaria, the craniofacial area that is relevant to this paper.
Finally, the last section of the paper is not a real conclusion, particularly the first paragraph (lines 611-618). This paragraph is a conclusion to the previous section (mechanotransduction in CS), not to the whole paper. This shows poor organization of the review paper.
Minor points:
Figure 2: it is impossible to read the names of the proteins in a hard copy of the file. I had to go to the pdf file and zoom in multiple times to be able to see the names. One more time this reflects the fact there is too much irrelevant information in the paper – the only proteins that matter are the ones in yellow (18 proteins). Similar to table 1, this figure should contain only the 18 genes relevant to the topic.
Be careful with the use of synonyms/similar words throughout the paper. Examples are: pathophysiology vs aetiology in the Abstract; calvarial sutures (line 32) vs cranial sutures (line 66); “fibers” (lines 113, 143) vs “fibres (line 121). Stick with one term.
Reviewer 2 Report
This is a comprehensive literature review with informative computational annotations on ciliary signaling in the pathophysiology of craniosynostosis. The schematic diagrams are impressive. The authors provide thoughtful insights into the role of primary cilia and mechanotransduction in CS pathogenesis. Please find my comments.
Table 1: “FGF signaling pathway” should be included in “Main pathways/molecular and biological functions” for FGFs and FGFRs.
Figure 2: The legend for the line thickness is not shown in the PDF file. If the colors of the nodes indicate categories or terms, a legend is required in the figure or they should be annotated in the supplemental table.
Line 331, proteins that failed to show significant interactions could still be important. Those genes may function relatively independently. Are there CS genes involved in ciliary signaling excluded by STRING analysis?
Line 397, (Rab23… hence serving as a negative regulator of the Hh signaling), Eggenschwiler et al. 2001 (PMID: 11449277) should be cited. There have been many new findings on the role of Rab23, e.g. Hasan et al. 2020 (PMID: 32662771) that the authors should review.
Line 428, which are the genes highlighted in Figure 1?
There are some typos, e.g. Line 338, “Figure 1” should be “Figure 2”; Figure 3, Line 414, “SMOH” should be “SMO”; Supplemental Table 5 is mislabeled as "Table S2” in the supplementary file.
Reviewer 3 Report
Dr. Tiberio and co-authors present a well written review of the current knowledge of the molecular genetics of craniosynostosis, the biology of primary cilia, and the observation of a potential overrepresentation of mutations in genes involved with ciliary biology in craniosynostosis.
The manuscript is well organized and provides an excellent overview on the topic.
Overall the arrangement of headings is good however, the writing is quite dense in section 3.1. This reviewer would suggest breaking down this section into subsections focusing on specific relationships between the ciliary genes and the craniosynostosis phenotypes associated with their mutations (e.g., synthesize the data in table 1 and figure 3).
Line 497-501: The authors state 13 of 18 genes are associated with sagittal, "more generally midline sutures". This statement would be much more powerful if a statistical statement of significance could be made. I would strongly suggest applying a statistical test to this observation. In addition, the use of the term "midline sutures" suggest a biologic or epidemiologic relationship for the pathogenesis of metopic and sagittal synostosis. Until there is strong biologic proof of this relationship, this reviewer would suggest merely stating the specific sutures associated with each type of synostosis.
Minor suggestion: Line 34: suggest substituting "fused" for "sealed together"
Figure 2 (as submitted for my review) is unreadable. If not supplied in a high resolution that can be easily viewed when published, this reviewer would suggest that it is included as a supplemental figure. An alternative would be to have the figure focus only on the networks of ciliary genes.
Round 2
Reviewer 1 Report
The authors addressed all major issues with the paper. The paper is now ready for publication.